# De-Suppression of Mesenchymal Cell Identities and Variable Phenotypic Outcomes Associated with Knockout of *Bbs1*

**DOI:** 10.3390/cells12222662

**Published:** 2023-11-20

**Authors:** Grace Mercedes Freke, Tiago Martins, Rosalind Jane Davies, Tina Beyer, Marian Seda, Emma Peskett, Naila Haq, Avishek Prasai, Georg Otto, Jeshmi Jeyabalan Srikaran, Victor Hernandez, Gaurav D. Diwan, Robert B. Russell, Marius Ueffing, Martina Huranova, Karsten Boldt, Philip L. Beales, Dagan Jenkins

**Affiliations:** 1Great Ormond Street Institute of Child Health, University College London, London WC1N 1EH, UK; grace.freke.15@ucl.ac.uk (G.M.F.); tiagofmendesmartins@gmail.com (T.M.); m.seda@ucl.ac.uk (M.S.); e.peskett@ucl.ac.uk (E.P.); naila.haq@kcl.ac.uk (N.H.); g.otto@imperial.ac.uk (G.O.); j.srikaran@ucl.ac.uk (J.J.S.); p.beales@ucl.ac.uk (P.L.B.); 2Institute for Ophthalmic Research, Center for Ophthalmology, University of Tübingen, Elfriede-Aulhorn-Strasse 7, 72076 Tübingen, Germany; tina.beyer@uni-tuebingen.de (T.B.); marius.ueffing@uni-tuebingen.de (M.U.); karsten.boldt@uni-tuebingen.de (K.B.); 3Laboratory of Adaptive Immunity, Institute of Molecular Genetics of the Czech Academy of Sciences, 14220 Prague, Czech Republicmartina.huranova@img.cas.cz (M.H.); 4Life Sciences Department, CHMLS, Brunel University London, Kingston Lane, Uxbridge UB8 3PH, UK; victor.hernandez@brunel.ac.uk; 5BioQuant, University of Heidelberg, Im Neuenheimer Feld 267, 69120 Heidelberg, Germany; gaurav.diwan@bioquant.uni-heidelberg.de (G.D.D.); robert.russell@bioquant.uni-heidelberg.de (R.B.R.); 6Biochemistry Center (BZH), University of Heidelberg, Im Neuenheimer Feld 328, 69120 Heidelberg, Germany

**Keywords:** Bardet–Biedl syndrome, primary cilia, epithelial-to-mesenchymal transition, kidney, collecting duct cells, Wnt signalling, fibrosis

## Abstract

Bardet–Biedl syndrome (BBS) is an archetypal ciliopathy caused by dysfunction of primary cilia. BBS affects multiple tissues, including the kidney, eye and hypothalamic satiety response. Understanding pan-tissue mechanisms of pathogenesis versus those which are tissue-specific, as well as gauging their associated inter-individual variation owing to genetic background and stochastic processes, is of paramount importance in syndromology. The BBSome is a membrane-trafficking and intraflagellar transport (IFT) adaptor protein complex formed by eight BBS proteins, including BBS1, which is the most commonly mutated gene in BBS. To investigate disease pathogenesis, we generated a series of clonal renal collecting duct IMCD3 cell lines carrying defined biallelic nonsense or frameshift mutations in *Bbs1*, as well as a panel of matching wild-type CRISPR control clones. Using a phenotypic screen and an unbiased multi-omics approach, we note significant clonal variability for all assays, emphasising the importance of analysing panels of genetically defined clones. Our results suggest that BBS1 is required for the suppression of mesenchymal cell identities as the IMCD3 cell passage number increases. This was associated with a failure to express epithelial cell markers and tight junction formation, which was variable amongst clones. Transcriptomic analysis of hypothalamic preparations from BBS mutant mice, as well as BBS patient fibroblasts, suggested that dysregulation of epithelial-to-mesenchymal transition (EMT) genes is a general predisposing feature of BBS across tissues. Collectively, this work suggests that the dynamic stability of the BBSome is essential for the suppression of mesenchymal cell identities as epithelial cells differentiate.

## 1. Introduction

Bardet–Biedl syndrome (BBS) is a phenotypically diverse condition, defined by a combination of major and minor clinical features including obesity, retinal degeneration and cystic kidney disease [1,2]. Over 20 causative genes have been identified in this autosomal recessive condition. *BBS1* is the most commonly mutated gene, and the p.M390R missense mutation in *BBS1* is the single most common mutation in BBS, accounting for ~30% of patients [3]. There is also significant variability in clinical presentation between patients with mutations in the same gene, suggesting that genetic background and allelic variation modulate mutational effects.

BBS is one of the founding diseases that originally defined the ciliopathies, caused by mutations in components of cilia. Many BBS proteins localise to the centrosomes and basal bodies, located at the base of cilia. This includes a subset of eight BBS proteins, including BBS1, that form the so-called BBSome protein complex which coats intracellular vesicles that traffic cargo to cilia [4,5]. It is unclear whether BBS proteins are required for ciliogenesis per se, and BBS proteins may have cilia-independent functions [5,6,7,8]. Contradictory results in this regard may relate to the use of a variety of cellular models, including transient gene knockdown in stable cell lines and analysis of heterogeneous cell types in primary cell culture.

Primary cilia are present on the surface of most cells in the body, where they act as hubs for a variety of signalling pathways. A key challenge is to dissect these varied roles of cilia in a multitude of cellular processes and to assign each of these roles to different disease phenotypes and mechanisms of pathogenesis. A difficulty here relates to the multiple molecular functions that individual ciliary proteins may simultaneously participate in and the possible interdependence of these functions. As such, holistic analyses of gene function are required.

Here, we have taken a systematic approach to determine the requirement for Bbs1 in renal epithelial cells and to identify correlates of phenotypic variability. We have used gene editing to generate a panel of inner medullary collecting duct (IMCD3) cells carrying defined biallelic frameshift mutations in *Bbs1*. These null-mutant cell lines have been analysed at the levels of gene copy number, whole-genome sequencing, transcriptomics and proteomics in comparison to matched clones that have been through the same CRISPR process, but which have a wild-type *Bbs1* sequence. We find that Bbs1 is dispensable for ciliogenesis but is required for the formation of the BBSome around the basal body. At the transcriptomic and proteomic levels, mesenchymal cell hallmark genes were consistently upregulated across clones, while epithelial genes were variably downregulated and correlated with *Bbs1* transcript instability and tight junction formation. The same epithelial-to-mesenchymal transition (EMT) signature was associated with BBSome loss-of-function in transcriptomic datasets from mouse hypothalamus and *BBS1* p.M390R mutant patient fibroblasts. Collectively, these results suggest that defects in EMT may be a generalised mechanism predisposing a variety of cell types to disease in BBS and lay the foundations for dissecting the underlying molecular mechanisms.

## 2. Results

### 2.1. Generation and Characterisation of Bbs1^-/-^ IMCD3 Cell Lines

Mouse and human *Bbs1*/*BBS1* genes show a remarkable degree of conservation in terms of transcript and protein sequence content and gene structure (Appendix A). In both species, the gene consists of 17 exons with a combined length of 1782 nucleotides encoding 593 amino acids. There is 87.6% identity between the human and mouse coding sequences and 92.2% identity between corresponding protein sequences.

Using an approach that we reported previously [9,10], we first generated biallelic frameshift mutations in *Bbs1* in clonal IMCD3 cell lines using gene editing. Three different sgRNAs were designed to target exon 1, 8 or 12 of the *Bbs1* gene (Appendix A). These gRNAs were selected as high-scoring sites that were considered likely to generate null alleles by targeting the translation initiation codon or being located in exons that would generate out-of-frame transcripts should they be skipped. Clones were genotyped by Sanger sequencing of the sgRNA target site, and those carrying biallelic null mutations were selected for further analysis. On average, 47.7% of clones carried biallelic indels which were presumptive knockouts for *Bbs1* (Appendix A). “WT-CRISPR” (WTcr) clones, which had been through the CRISPR/Cas9 gene-editing pipeline but had remained wild-type (WT), were also identified and maintained as clonal controls (Appendix A).

Using Western blotting, we identified nine clones that were *Bbs1* “knock-outs” (*Bbs1*^-/-^), demonstrating a complete absence of detectable Bbs1 protein, as well as a clone carrying the p.M390R mutation which was generated using a homology-directed repair template (Appendix A, Figure 1A; genotypes of individual clones are listed, Appendix A). Quantification of *Bbs1* transcript levels by qPCR demonstrated clonal variation in expression amongst WTcr clones and a variable reduction in *Bbs1* transcript levels in clones carrying biallelic indels in the gene, which presumably resulted from nonsense-mediated mRNA decay (NMD) (Figure 1B).

Given that some instances of aneuploidy have been observed in IMCD3 cells, we confirmed a *Bbs1* gene copy number of 2 by genomic qPCR in all but one clone (Bbs1^ex12.2^). We also subjected all eight *Bbs1*^-/-^ clones to whole-genome sequencing (WGS). A total of 157,791 variants in 15,219 genes were found within coding and splice regions (7 bp on either side of the coding regions) of the eight clones. Within *Bbs1*, a silent SNP (NC_000085.6:g.4984905C>G) was found to be heterozygous in all clones and was also confirmed in the WTcr clones. No homozygous or compound heterozygous mutations were found in a comprehensive panel of 1493 cilia genes compiled from the SysCilia gold standard of ciliary genes [11], CiliaCarta Bayesian predictions [12] or Gene Ontology annotated genes [13,14].

### 2.2. Characterisation of Ciliogenesis in Bbs1^-/-^ IMCD3 Cell Lines

We began our phenotypic analyses by investigating whether *Bbs1*^-/-^ IMCD3 clones recapitulated expected or known BBS1-associated phenotypes. All but one of the *Bbs1*^-/-^ clones formed cilia, and these cilia had the expected localisation and distribution of acetylated α-tubulin, γ-tubulin and ARL13B (Figure 1C–G). Generally, there was no consistent reduction in the percentage of ciliated cells for each clone, compared to either its respective WTcr control or the parental WT IMCD3 cell line. However, one *Bbs1*^-/-^ clone and one WTcr clone did not form cilia, although normal cell density was observed (Figure 1D–G). This suggested that there may be clonal differences in ciliogenic potential. Analysis of ciliogenesis in clonal cell lines isolated from the parental WT cell line (without gene editing) demonstrated that 7.9% (i.e., 5 out of 63) of IMCD3 cell clones are unable to form cilia, which was not significantly different to the proportion of *Bbs1*^-/-^ clones that completely lacked cilia (Fisher’s exact test, *p* = 0.3792). We conclude that Bbs1 is not essential for the formation of cilia in IMCD3 cells.

### 2.3. Impact of Loss of Bbs1 on the BBSome

The BBSome is a complex of eight proteins, including BBS1, which associate with basal bodies and pericentriolar satellites [1,3,15,16]. Prasai et al. [16] showed that a lack of BBS1 results in retention of the remaining seven BBSome subunits at the pericentriolar satellites in the form of an unstable pre-BBSome complex. Thus, the localisation of BBSome subunits to the pericentriolar satellites (PCM1) represents the preliminary step that precedes the final assembly and transfer of the BBSome to the cilia. We generated retroviruses that express mCherry-BBS1 or YFP-BBS8 in order to study BBSome function and integrity in Bbs1^-/-^ clones. PCM1 (pericentriolar material-1) staining showed that pericentriolar satellites cluster tightly around the basal bodies in IMCD3 cells that were grown to confluence and serum-starved (Figure 2A). YFP-BBS1 and YFP-BBS8 showed only very limited localisation with PCM-1 in WTcr IMCD3 cells (Figure 2A), as observed previously in RPE-1 cells [16]. By contrast, co-localisation of YFP-BBS8 with PCM-1 increased significantly in two Bbs1^-/-^ cell clones (Figure 2A), indicating accumulation of the incomplete pre-BBSome at the pericentriolar satellites [16]. This phenotype was rescued by expression of mCherry-BBS1 in Bbs1^-/-^ cells (Figure 2A), confirming the specificity of these findings and validating these cell lines as a faithful model of Bbs1 loss of function.

p.M390R is a hypomorphic mutation in BBS1 and sits at the binding interface between BBS1 and ARL6, the small GTPase that regulates BBSome recycling [4]. To investigate BBSome complex formation, we used tandem affinity purification using either the wild-type or p.M390R mutant version of BBS1 as bait. Wild-type BBS1 was able to pull down BBSome components BBS2, BBS4, BBS5, BBS9 and TTC8, and these interactions were significantly enriched over an irrelevant control bait protein (Figure 2B). By contrast, p.M390R BBS1 exhibited a specific and complete inability to retrieve BBS4 and BBS5 in these experiments, although binding to other BBSome components was quantitatively unaffected. Interaction of wild-type BBS1 with BBS4 and BBS5 was enriched to the same degree when compared to either the p.M390R or control bait proteins, indicating that interaction of p.M390R BBS1 with BBS4/5 was completely abrogated (Figure 2B). Consistent with these findings, structural resolution of the BBSome has revealed that BBS1 interacts directly with BBS4, with a conformation that is dependent on ARL6(GTP) or ARL6(GDP) binding [17]. Collectively, these data show that BBS1 is required for BBSome complex formation and/or turnover, and its loss leads to accumulation of the BBSome at pericentriolar satellites.

### 2.4. A Phenotypic Screen Reveals Clonal Variation in Cell–Cell Junction Formation in Bbs1^-/-^ Clones

BBSome components have been linked to a number of different cellular processes under both ciliogenic and non-ciliogenic conditions [1]. We therefore undertook a phenotypic screen to evaluate the requirement for Bbs1 in these processes (Appendix A). First, using the Bbs1^ex8^ clone, we tested for differences in midbody formation, cell viability, population doubling time, programmed cell death and DNA damage response, and no differences to wild type were observed. Another cellular process linked to the BBSome is cell migration. In scratch wound assays, statistically significant clonal differences in cell migration were observed within the WTcr and *Bbs1*^-/-^ groups of clones, but no consistent differences were observed between Wtcr and *Bbs1*^-/-^ clones. Cell density at confluence (number of nuclei per field of view) and saturation density were increased in this clone (Figure 1E and Appendix A). Phalloidin staining of F-actin suggested aberrant formation of cell–cell junctions in Bbs1^ex8^ mutant cells, indicated by less clearly defined cell boundaries (Figure 3). Gaps were seen between Bbs1^ex8^ cells, despite cells being confluent, and there was reduced β-catenin at cell peripheries.

Cell–cell junctions, such as tight junctions and adherens junctions, play a role in signalling to limit proliferation once adjacent cells come into contact with one another [18,19]. All but one of the WTcr cell lines (WTcr^ex8^) formed both adherens junctions (E-cadherin, β-catenin) and tight junctions (ZO-1, occludin). There was a spectrum of variation in the ability of *Bbs1*^-/-^ clones to form cell–cell junctions: few or no Bbs1^ex8^ or Bbs1^ex1.1^ cells formed adherens or tight junctions, while Bbs1^ex12.4^ cells formed only isolated patches of adherens or tight junctions (Figure 3 and Figure 4). Therefore, the ability of cells to form tight and adherens junctions varied between *Bbs1*^-/-^ clones but was generally consistent amongst WTcr clones.

### 2.5. Enrichment of Mesenchymal Cell Identities in Bbs1^-/-^ Clones Compared to Passage-Matched Controls

RNA sequencing (RNAseq) was used to identify cellular processes affected by loss of BBS1 across all *Bbs1*^-/-^ clones. There was a close correspondence between relative abundance of *Bbs1* mRNA as determined by qPCR and RNAseq (Figure 1B and Appendix A). Of the known BBS disease genes, three (*Bbs2*, *Mks1/Bbs13*, *Wdpcp/Bbs15*) were consistently upregulated in *Bbs1*^-/-^ clones, with *Mks1* showing a ~12-fold upregulation (Appendix A). Globally, 2077 upregulated and 902 downregulated differentially expressed genes (DEGs; log_2_ fold change > 1.00, *p*-adj < 0.05) were identified as being common to all *Bbs1*^-/-^ clones.

Gene ontology (GO) term enrichment analysis indicated significant enrichment of genes associated with the *extracellular matrix (ECM), Wnt and notch signalling, metallopeptidases* and *growth factor binding* (Figure 5A). Closer inspection showed interconnectivity between the DEGs contributing to enrichment of each of these terms, which all included ECM components (Appendix A).

Hallmark gene set enrichment analysis [20,21,22], which uses the entire transcriptomic dataset, was implemented as a complementary approach. Given the extensive phenotypic clonal variation that was observed across *Bbs1*^-/-^ clones (described above), GSEA was performed for the comparison of each individual knockout clone to its passage-matched WTcr, in addition to group-level analysis (*Bbs1*^-/-^ versus WT groups). GSEA indicated some clonal differences, as well as several common signatures in the transcriptomes of the six *Bbs1*^-/-^ clones (Figure 5A). Amongst significantly enriched gene sets were *Wnt beta-catenin signalling*, *notch signalling, myogenesis* and *epithelial–mesenchymal transition*. At the clone level, only *epithelial–mesenchymal transition* and *myogenesis* were positively enriched in all six *Bbs1*^-/-^ clones, while the *Wnt beta-catenin signalling* gene set was positively enriched in all but one *Bbs1*^-/-^ clone.

To complement RNAseq datasets, we assessed differences in protein abundance in whole-cell lysates of two *Bbs1*^-/-^ clones, Bbs1^ex1.4^ and Bbs1^ex8^, by mass spectrometry, as these clones demonstrated discordant tight junction phenotypes. GSEA demonstrated that only the *epithelial–mesenchymal transition* (EMT) hallmark gene set was common to both the transcriptomic and proteomic datasets (Appendix A). Collectively, these data further demonstrate clonal variation in *Bbs1*^-/-^ clones and led us to focus on the EMT phenotype as a generalised feature of *Bbs1* knockout.

### 2.6. EMT Gene Set Enrichment Is a Feature of Diverse BBSome Transcriptomic Datasets

We next looked for commonly enriched gene sets in several other of our own and published [23] datasets (Figure 5B). These datasets were derived from loss-of-function models of BBS1 and other BBSome genes, which we assumed might share transcriptional changes underlying the loss of the BBSome. This included a panel of primary human fibroblast monolayers from patients who were homozygous for the p.M390R mutation in *BBS1*, which we had collected as part of the HipSci project [24] and had been grown under ciliogenic conditions. We also analysed adult mouse hypothalamic tissue from the p.M390R *Bbs1* and *Bbs4* knockout mice. Finally, we additionally analysed a published dataset from HeLa cells in which BBS7 had been knocked down using RNAi [23]. The only hallmark gene set that was enriched across this diverse dataset was EMT, suggesting a generalised role in diverse cell and tissue types.

### 2.7. Confirmation of EMT DEGs and Analysis of Clonal Variation

The hallmark *epithelial–mesenchymal transition* gene set contains 200 mesenchymal biomarkers. As no hallmark gene set exists for epithelial markers, or for mesenchymal–epithelial transition (MET), a list of 131 core genes from published EMT gene expression studies was used [25]. The list comprised 68 genes that are upregulated in EMT (“Gröger” mesenchymal genes) and 63 that are downregulated in EMT (“Gröger” epithelial genes). We performed a group-level GSEA of DEGs in *Bbs1*^-/-^ clones using these two custom-made gene sets. There was significant positive enrichment of the Gröger mesenchymal gene set, which had a normalised enrichment score (NES) of 2.10 (nominal *p*-value = 0.000, FDR = 0.000). By contrast, the Gröger epithelial gene set had a non-significant NES of 1.08 (nominal *p*-value = 0.363, FDR = 0.355). Clone-level inspection of log_2_ fold-change values of genes within each set indicated that many Gröger mesenchymal genes (including *Nid2*, *Vim*, *Pmp22*, *Fgfr1*, *Wnt5a*, *Zeb1*, *Cdh2*) were significantly upregulated in all knockout clones, and no single mesenchymal gene was significantly downregulated in all *Bbs1*^-/-^ samples. We confirmed this general trend for a panel of eleven Groger EMT genes in the six *Bbs1*^-/-^ clones using RT-qPCR (Figure 5C) and, to control for clonal variation, RT-qPCR was also performed on a set of six WTcr clones (Figure 5D). There was a statistically significant difference in the mean expression level of each gene between the group of *Bbs1*^-/-^ clones and the group of WTcr clones (*p* = 0.00008033 by two-tailed t-test, paired by gene).

In some clones, we observed a general trend toward downregulation of epithelial genes (e.g., *Bbs1^ex8^*, 19/46 epithelial vs. 1533/15,205 non-epithelial genes downregulated, *p <* 0.0001), and we noted a significant downregulation of several of the Gröger epithelial genes in all six knockout clones. Therefore, we conclude that mesenchymal genes were consistently upregulated across all *Bbs1*^-/-^ clones, while there was clonal variation with respect to the downregulation of epithelial markers.

### 2.8. Suppression of Mesenchymal Cell Identities Is a Feature of Renal Epithelial Cell Maturation

Unsupervised clustering analysis and principal component analysis (PCA) of the RNAseq datasets demonstrated that *Bbs1*^-/-^ samples clustered separately from passage-matched wild-type samples. The exception to this was an early passage (P10) wild-type parental cell line sourced from ATCC (‘WT-P10), which clustered with the *Bbs1*^-/-^ clones. Inspection of RNAseq reads aligned to the mouse *Bbs1* gene sequence (using Integrative Genomics Viewer, IGV) confirmed the correct genotypes. RT-qPCR confirmed that mesenchymal genes were upregulated in WT-P10 relative to later-passage WT and WTcr cell lines, to a similar degree as observed for *Bbs1*^-/-^ clones.

To investigate the relationship between the IMCD3 cell passage number and mesenchymal cell marker gene expression, four independent investigators aged WT-P10 cells until passage number 64. At passage numbers 10, 12, 22/23/24, 32/33/34, 42/43/44, 52/53/54 and 62/63/64, samples of these cells were grown under ciliogenic conditions. RT-qPCR demonstrated a general trend whereby mesenchymal gene expression was downregulated as the cell passage number increased (Figure 6). Therefore, late-passage *Bbs1*^-/-^ clones exhibit a phenotype that resembles early passage parental wild-type cells, suggesting that BBS1 is required for suppression of mesenchymal cell identities as renal epithelial cells mature.

## 3. Discussion

Here, we have attempted to take a relatively unbiased approach to understand the cellular phenotypes associated with a complete loss of Bbs1 protein. We also wanted to capture the biological variability associated with these phenotypes and to avoid technical variation associated with gene knockdown. To this end, we generated a panel of clonal IMCD3 cell lines carrying defined biallelic mutations in *Bbs1*, and we confirmed that all of the clones under study generated no detectable Bbs1 proteins. Other potential confounders relate to off-target effects of gene editing and genomic instability, as well as as-yet-undescribed artefacts of the CRISPR/Cas9 system. We controlled for these factors by WGS of all of our clones and quantification of DNA copy number at the *Bbs1* locus. In our analyses, we also included a panel of matched WTcr clones which had been through the same gene-editing experiment as their matched *Bbs1* mutant but had remained wild type for *Bbs1*. WGS demonstrated that off-target mutations are not common using the CRISPR/Cas9 system, as reported previously [9,10,26]. This panel of *Bbs1* mutant cell clones therefore represents a valuable resource to interrogate the cellular requirements for this gene.

By screening a variety of cellular phenotypes that have previously been associated with ciliopathies, we noted a significant degree of clonal variation. There have been mixed reports linking BBS1 to ciliogenesis; however, we found that gross ciliary anatomy was normal in all but one of our *Bbs1*^-/-^ clones. This proportional loss of ciliogenic potential was mirrored in parental IMCD3 cells where 7.9% of cellular clones failed to form cilia. We also noted prominent tight junction defects in three *Bbs1*^-/-^ clones. Therefore, clonal variability in phenotypic outcomes is an important consideration, and we could easily have reached very different conclusions if we had either focused on just a few clones or if we had not taken a clonal analysis, where phenotypes may be masked in heterogeneous populations of cells.

The major conclusions of this work are that: (1) Loss of Bbs1 is associated with differential expression of EMT hallmark genes, in particular representing enrichment for mesenchymal cell identities and variable loss of epithelial marker gene expression. (2) This appeared to represent a failure to downregulate mesenchymal cell identities as cells aged through subsequent passages, as opposed to any kind of dedifferentiation. (3) This did not correlate with morphological defects in ciliogenesis.

Bulk transcriptomic analyses revealed that differentially expressed genes were significantly enriched for EMT genes in all *Bbs1*^-/-^ clones analysed. We note that EMT has not previously been linked to BBS, although EMT-like changes have been identified in models of other ciliopathy genes. Kidney organoids grown from patient-derived iPSCs carrying *IFT140* mutations displayed a downregulation of genes involved in apical–basal polarity and cell–cell junctions [27], and incomplete formation of tight junctions has been observed in *Bbs8* models [28]. Activation of RhoA is associated with EMT [29,30], which might suggest that increased RhoA activity reported in *Bbs4^-/-^ and Bbs6^-/-^* mouse embryonic fibroblasts was due to EMT-like transcriptional changes [6]. Like *Bbs1*, ciliary transition zone components NPHP1 and NPHP4 are upregulated during epithelial cell polarisation, and their depletion has been shown to disrupt tight junction formation [31,32]. NPHP1 and NPHP4 localise cell–cell junctions and interact with PALS1, PAR3 and PATJ, which are components of the epithelial polarity program [31,32].

While mesenchymal gene upregulation was quite uniform across *Bbs1*^-/-^ clones, there was a significant degree of variation amongst WTcr clones. This is somewhat counterintuitive, since phenotypic variability is usually higher amongst mutants where robustness conferred by network topology can be disrupted. We found that early passage parental IMCD3 cells demonstrated similar mesenchymal gene expression levels as *Bbs1*^-/-^ clones, i.e., these genes were upregulated in early passage parental wild types as compared to WTcr clones that were, necessarily, of a later passage number. We found that mesenchymal gene expression in parental wild-type IMCD3 cells was progressively downregulated over the course of >60 passages. This suggests that Bbs1 is required for these epithelial cells to downregulate mesenchymal genes as they mature.

Enrichment for EMT genes was found in a number of cell lines in addition to IMCD3 cells, including human *BBS1* mutant fibroblasts and *Bbs4/6* mutant mouse hypothalamus. One question that emerges in relation to BBS pathogenesis is how disruption of a potentially ubiquitous phenomenon (if this turns out to be the case) can generate a syndrome in which a specific set of tissues are affected? While this requires future investigation, there are at least two possible explanations. Firstly, there may be a greater demand for EMT processes in certain tissues. For instance, BBS has been shown to regulate neural crest migration, which involves EMT and delamination of neuroepithelial cells [33]. An analogous situation has been described in Treacher Collins and Miller syndromes where mutations affecting the ubiquitous processes of ribosome biogenesis and pyrimidine biosynthesis cause very specific neurocristopathic clinical features [34,35,36,37].

A second explanation may relate to the aging process. Different epithelial structures may require a different degree of maturity to function, or they may have a greater turnover owing to environmental exposures or high rates of proteostasis. Such a scenario might be relevant to photoreceptors within the eye. EMT has been found to play a crucial role in the age-related development of fibrosis in the heart and lungs [38], and fibrosis is seen in the kidneys and hepato-biliary systems of some BBS patients [9,39,40,41,42,43,44,45,46].

How can we explain phenotypic variation between clones? Clonal variation could relate to genetic or epigenetic differences, or heterogeneity of cellular state, which could cause cells to respond differently to the homogenous experimental conditions in which they were grown. Our WGS did indeed demonstrate genetic differences between clones, but no mutations were shared amongst clones, nor were they present in homozygous form in ciliary genes. (We did identify an SNP in *Bbs1* present in all clones and the parental cells.) In terms of epigenetics, we would expect to see these differences reflected in transcriptional changes. We identified discordant gene expression changes when a directed analysis of the severe ex8 and mild ex1.4 clones were compared. However, a wider study will be required to follow up on any relevance of these differences to clonal variation. The finding that a significant proportion of IMCD3 cell clones never form cilia, whereas a typical rate of ciliation of approximately 60% occurs in all other clones, does indeed suggest that different cellular states may be present. Once again, a systematic comparison of panels of wild-type clones with and without ciliogenic potential would need to be undertaken. In our work, for the purpose of rigour, we focused on the EMT phenotype that was consistent across clones.

We did note that cell clones with most severe phenotypes (ex8, ex1.1) demonstrated resistance to *Bbs1* transcript instability (i.e., higher transcript abundance) than other clones. In quantitative transepithelial electrical resistance (TEER) assays, which is a functional assessment of tight junctions, a strong inverse correlation between Bbs1 transcript abundance and TEER was observed (Appendix A). We considered that this might reflect a form of genetic compensation, whereby transcript instability is detected by an as-yet-uncharacterised mechanism and leads to the upregulation of functionally paralogous genes. However, we were unable to detect such a candidate gene, and this interesting possibility will require future investigation.

Visual inspection of the GSEA heatmap data does support a consistent transcriptional signature across all *Bbs1*^-/-^ clones, representing at least 16 gene sets. Although not all of these were statistically significant in all clones, this could reflect statistical power, and so we have certainly focused on the most high-confidence findings. The whole cell proteomics of just two discordant clones also showed that a number of gene sets were common to both, and EMT was once again enriched in both clones. Therefore, there may be Bbs1-independent sources of clonal variation, factors of variation that interact with Bbs1 and also differences in cell state, as discussed above.

A number of statistically significant DEGs encoding signalling pathway components were identified in all *Bbs1*^-/-^ clones that have previously been linked to cilia and could be mechanistically related to EMT. In Figure 5A,B, *mTORC1*, *TNFA* and *MYC targets* are negatively enriched in both transcriptomics and proteomic datasets, and *WNT/β-catenin*, *pancreas/β-cells* and *myogenesis* gene sets are positively enriched in all but one clone. *WNT/β-catenin* and *MYC* have been linked to EMT and fibrosis and so could be valuable targets to investigate in future work. In future, it will be necessary to delineate the gene regulatory network linking BBS1 to EMT.

## 4. Materials and Methods

### 4.1. Ethics

The study was ethically approved by the London Bloomsbury Research Ethics Committee 524 (08/H0713/82) and conducted at the Great Ormond Street Institute of Child Health. Patient fibroblasts were obtained from affected individuals and healthy volunteers under ethical approval granted through the Health Research Authority London Bloomsbury Research Ethics Committee (REC 08/H0713/82; IRAS 103488).

### 4.2. Statistical Tools

Statistical analyses were performed using GraphPad Prism version 10.0.0 for Windows, GraphPad Software, Boston, MA, USA, www.graphpad.com.

### 4.3. Mass Spectrometry

Protein complex purification, sample preparation and mass spectrometry-based analysis were performed as described before [47,48,49]. In brief, N-terminal Strep/FLAG-tagged human BBS1 wild type (ENST00000318312.12) and M390R mutant constructs were transfected, using homemade polyethylenimine solution, into HEK293T cells grown in 14 cm plates (Dulbecco’s Modified Eagle Medium supplemented with 10% FBS and 0.5% penicillin/streptomycin), respectively. C-terminal Strep/FLAG-tagged RAF1 was used as control. Forty-eight hours after transfection, the cells were washed with DPBS and lysed (1× TBS, 0.5% Nonident P40 (Roche), 1% phosphatase inhibitor cocktail 2 and 3 (Sigma, Tohoku, Japan) and 2% protease inhibitor cocktail (Roche, Vaud, Switzerland)). The incubation at 4 °C for 30 min in an end-over-end shaker was followed by centrifugation with 10,000× *g* at 4 °C. The supernatant was used for Strep-based immunoprecipitation. The lysate was incubated with beads for 90 min. Bound protein/protein complexes were washed 3 times (1× TBS, 0.5% Nonident P40 (Roche, Vaud, Switzerland), 1% phosphatase inhibitor cocktail 2 and 3 (Sigma, Tohoku, Japan)), followed by elution using Strep elution buffer. Protein precipitation using chloroform and methanol, trypsin digestion and peptide purification was performed as described before [47,48,49].

MS/MS data were analysed using MaxQuant (version 1.6.1.0) [50] and the SwissProt human database (2019_11, #20,367 entries). Trypsin/P was selected as digesting enzyme, and cysteine carbamidomethylation was set as fixed modification. Data were analysed using label-free quantification with a minimum ratio count of 2, first search peptide tolerance of 20, minimum peptide length of 7 and a minimum peptide number of 1. Razor and unique peptides were used for protein quantification. The match between run options was enabled with a match time window of 0.7 min and an alignment time window of 20 min.

The statistical analysis was performed using the Perseus software (version 1.6.5.0) [51]. In total, 9 biological replicates prepared in two separate experiments were analysed. The data were filtered for “contaminants” (based on the MaxQuant contaminant list), peptides “Only identified by side” or “Reverse” sequences and a minimum of 5 valid values in at least one group (control, BBS1 WT or BBS1 M390R). Based on the mean value, significant outliers between groups were detected using the significance A (Benjamini–Hochberg FDR) < 0.05. The stability of protein enrichment between samples was determined using permutation-based FDR < 0.05. Only proteins, which passed both tests for control versus BBS1 WT or control versus BBS1 M390R, as well as BBS1 WT versus BBS1 M390R, were considered as being significantly changed in the mutant.

### 4.4. Flow Cytometry

IMCD3 cells expressing the YFP-tagged subunits were cultivated in 24-well dishes, trypsinized and centrifuged at 1000× *g* for 1 min. The pellets were washed in PBS and resuspended in FACS buffer (2 mm EDTA, 2% FBS and 0.1% sodium azide). Measurements were taken on Aurora™ (Cytek Biosciences, Fremont, CA, USA). Geometric means of fluorescence intensity of YFP-positive cells were obtained and used to examine the protein expression. Flow cytometry data were analysed using FlowJo v10.9 (BD Biosciences, NJ, USA).

### 4.5. RNA Extraction for RNA-Seq

In total, 450,000 cells/well were seeded in a 6-well plate in complete media. Twenty-four hours post-seeding, cells were washed three times with PBS and serum-starved for 48 h in low serum media. Cells were washed three times with ice-cold PBS and lysed at room temperature with 1 mL/well QIAzol lysis reagent (QIAGEN). Lysates were incubated at room temperature for 5 min and homogenised by pipetting up and down. They were then transferred to 1.5 mL microcentrifuge tubes and stored at −80 °C until RNA isolation. To isolate RNA, 0.2 mL chloroform was added. Tubes were securely capped and incubated at room temperature for 3 min. Samples were centrifuged at 12,000× *g* at 4 °C for 15 min. The aqueous phase of each sample (containing RNA) was transferred to a 1.5 mL microcentrifuge tube, and an equal volume of 100% ethanol was added. Samples were mixed well. The Monarch Total RNA Miniprep kit (NEB) was then used for gDNA elimination and final purification of RNA. The kit contained RNA purification columns, collection tubes, RNA wash buffer, DNase I, DNase I reaction buffer and RNA priming buffer. Then, 100 mL of 100% ethanol was added to the 25 mL RNA wash buffer concentrate supplied in the kit. All subsequent centrifugation steps were performed at 16,000× *g*. Samples were transferred to RNA purification columns fitted with collection tubes. Tubes were spun for 30 s, and flow-through was discarded. On-column DNase I treatment was performed to eliminate gDNA, as follows. Columns were washed with 500 µL RNA wash buffer and spun for 30 s. Flow-through was discarded. In a separate 1.5 mL microcentrifuge tube, 5 µL DNase I was mixed with 75 µL DNase I reaction buffer and added directly to the RNA purification column. Samples were incubated for 15 min at room temperature. Then, 500 µL of RNA priming buffer was added to columns, and columns were spun for 30 s. Flow-through was discarded. Then, 500 µL of RNA wash buffer was added, columns were spun for 30 s and flow-through was discarded. Next, 500 µL of RNA wash buffer was added again, and columns were spun for 2 min. The column was transferred to a clean 1.5 mL microcentrifuge tube, avoiding contact with the flow-through in the collection tube. RNA was eluted in 50 µL of RNase-free H_2_O. RNA concentration and purity were assayed by NanoDrop 1000 (Thermo Fisher Scientific). A 260/280 ratio of 2.0 and a 260/230 ratio of 1.8–2.2 were accepted as pure for RNA.

### 4.6. RNA Sequencing and Alignment

Samples were anonymised using alphanumeric nomenclature (replicates “1”, “2” or “3” from cell lines “A”, “B”, “C”, etc.) to avoid biasing of RNA-seq data analysis. An amount of 2 µg each of total RNA sample was allocated for RNA-seq. Library preparation and sequencing were performed by GENEWIZ LLC (South Plainfield, NJ, USA). GENEWIZ described its methodologies as follows:

RNA samples were quantified with a Qubit 2.0 Fluorometer (Life Technologies, Carlsbad, CA, USA), and RNA integrity was checked with 4200 TapeStation (Agilent Technologies, Santa Clara, CA, USA).

RNA sequencing library preparation used the NEBNext Ultra RNA Library Prep Kit for Illumina (NEB), following manufacturer’s instructions. Briefly, mRNAs were first enriched with Oligo-d(T) beads. Enriched mRNAs were fragmented for 15 min at 94 °C. First-strand and second-strand cDNA were subsequently synthesized. cDNA fragments were end-repaired and adenylated at 3’ ends, and universal adapters were ligated to cDNA fragments, followed by index addition and library enrichment by PCR with limited cycles. The sequencing library was validated on the Agilent TapeStation (Agilent Technologies) and quantified by using Qubit 2.0 Fluorometer (Invitrogen, Waltham, MA, USA), as well as by quantitative PCR (KAPA Biosystems, Wilmington, MA, USA).

The sequencing libraries were clustered on a single lane of a flowcell. After clustering, the flowcell was loaded on the Illumina HiSeq instrument according to manufacturer’s instructions. The samples were sequenced using a 2 × 150 paired-end configuration. Image analysis and base calling were conducted by the HiSeq Control Software (Illumina, San Diego, CA, USA). Raw sequence data (BCL files) generated from Illumina HiSeq were converted into Fastq files and demultiplexed using Illumina’s bcl2fastq 2.17 software. One mismatch was allowed for index sequence identification.

Sequence quality was evaluated. Sequence reads were trimmed to remove possible adapter sequences and nucleotides with poor quality using Trimmomatic v0.36 [52]. The trimmed reads were mapped to the mouse GRCm38/mm10 reference genome using the STAR aligner v2.5.2b [53]. The STAR aligner is a splice aligner that detects splice junctions and incorporates them to help align entire read sequences. BAM files were generated from this step. Unique gene hit counts were calculated by using feature Counts from the Subread package v1.5.2 [54]. Only unique reads that fell within exon regions were counted. Since a strand-specific library preparation was performed, the reads were strand-specifically counted.

### 4.7. Differential Gene Expression Analysis

Gene hit counts were used for downstream data quality control and differential gene expression analysis with DESeq2 [55]. Unsupervised clustering analysis and prinicipal component analysis (PCA) were used to identify and exclude outlier samples from downstream analyses and to confirm absence of batch effects between 2019 and 2021 datasets.

There were two approaches to differential gene expression analysis: (i) in a group-level approach, gene expression in the group of *Bbs1*^-/-^ clones (excluding Bbs1^ex12.1^, for the core analysis) was compared to that in the group of WT samples (excluding WT P.10, for the core analysis); and (ii) in a clone-level approach, gene expression in each individual *Bbs1*^-/-^ clone was compared to that in a passage-matched WT sample.

Group-level differential gene expression analysis was conducted in-house. Clone-level differential gene expression analysis was conducted by GENEWIZ LLC (South Plainfield, NJ, USA). In DESeq2, a comparison of gene hit counts between the knockout and WT groups (for group-level analysis), or between each knockout clone and its passage-matched WT (for the clone-level analysis), was performed. *p*-values and log_2_ fold changes were generated using the Wald test. For each comparison, genes with an adjusted *p*-value < 0.05 and absolute log_2_ fold change > 1.00 were called differentially expressed genes (DEGs).

### 4.8. Gene Set Enrichment Analysis Using g:profiler

Gene set enrichment analysis (GSEA) was performed using g:profiler [56] to identify over-represented gene ontology (GO) terms associated with differentially expressed genes (DEGs) from the group-level approach. A g:GOSt query was run in a web browser application, through the online g:profiler interface (https://biit.cs.ut.ee/gprofiler/gost, accessed on February 2021). Separate queries were run for lists of up- and downregulated DEGs (output from differential gene expression analysis). Ensembl identifiers of genes, ordered by decreasing absolute log_2_ fold change, were input into the query input box and the “ordered query” tick box was checked. Each query was run using default settings. Results were exported as portable network graphic (PNG) image files.

### 4.9. Gene Set Enrichment Analysis Using the GSEA Application

Gene set enrichment analysis [20] was performed using the desktop GSEA software v2023.2 application, with hallmark [22] or custom-made Gröger epithelial and mesenchymal [25] gene sets. Hallmark gene sets were downloaded from the online Molecular Signatures Database (MSigDB) [20,21] as gene matrix transposed (GMT) files containing gene symbols. A GMT file was made for the Gröger epithelial and mesenchymal gene sets in Microsoft Excel, in such a manner that each row of the worksheet represented a different gene set. Gene set name was entered into the first column, a gene set description into the second column and cells in subsequent columns contained (capitalised) gene symbols of the genes within the set. The worksheet was saved as a tab-delimited file with the extension .gmt.

For each comparison (the group of *Bbs1*^-/-^ clones vs. the group of WTcr clones, for the group-level comparison, and each *Bbs1*^-/-^ clone vs. its passage-matched WTcr, for each clone level comparison), an RNK file was generated, as follows. In Microsoft Excel, a table was created containing the complete list of sequenced gene symbols in the first column and the respective log_2_ fold change in its expression in the second column. Gene symbols were fully capitalised, to match the nomenclature used in the gene set GMT files. The table was ranked from largest to smallest log_2_ fold change. The worksheet was saved as a tab- delimited file with the extension .rnk.

RNK and GMT files were loaded into the GSEA application. GSEA Preranked analysis was performed for each combination of RNK file (i.e., comparison) and GMT file (i.e., gene set), using default settings. A gene set was considered significantly enriched if its normalised enrichment score (NES) was associated with a nominal *p*-value < 0.05 and a false discovery rate (FDR) < 0.25 [20]. A heatmap of results was generated using the R programming language in the R-Studio application, with the package *pheatmaps*. The heatmap was coloured by normalised enrichment score (NES) and annotated to show gene sets that were significantly positively (“+”) or significantly negatively (“−”) enriched (i.e., with a nominal *p*-value < 0.05 and a false discovery rate (FDR) < 0.25).

### 4.10. Cell Culture

Phoenix-Ampho cells were a kind gift from Dr. Thomas Brdicka (Institute of Molecular Genetics of the Czech Academy of Sciences). Cells were cultured in complete Dulbecco’s Modified Eagle’s Media (DMEM) (Sigma, D6429-500 mL) supplemented with 10% FBS (Gibco, Grand Island, NY, USA, 10270-106), 100 units/mL of penicillin (BB Pharma, Lhotkam Cheque Republic), 100 μg/mL of streptomycin (Sigma Aldrich) and 40 μg/mL of gentamicin (Sandoz) at 37 °C and 5% CO_2_.

### 4.11. Antibodies

Mouse antibody against PCM-1 (sc-398365) was purchased from Santa Cruz Biotechnology. Rabbit antibody against GFP (A11122) was purchased from Invitrogen.

Secondary antibodies were as follows: anti-rabbit Alexa Fluor 488 (Invitrogen, A11008; 1:1000) and anti-mouse Alexa Fluor 647 (Invitrogen, A21235; 1:1000).

### 4.12. Virus and Cell Line Preparation

Cloning of YFP-BBS8 is described in Prasai et al., 2020 [16]. BBS1 was fused to mCherry N-terminally using recombinant PCR and cloned into pMSCV-Thy-IRES 1.1 vector (Clontech, Mountain View, CA, USA) using EcoRI and ClaI restriction sites.

In total, 30 μg of plasmid DNA was transfected into Phoenix-Ampho cells using the polyethylenimine to generate YFP-BBS8 and mCherry-BBS1 retroviruses. IMCD3 cells were transduced using 2 mL of virus aliquot in the presence of 8 µg/mL of polybrene using standard protocols. IMCD3 cells stably expressing YFP-BBS8 and mCherry-BBS1 were sorted using the 488 nm and 561 nm laser on FACSAria IIu (BD Biosciences, New Jersey, USA).

### 4.13. Immunofluorescence

IMCD3 cells were cultured on 12 mm coverslips and serum-starved for 24 h. Cells were fixed (4% formaldehyde) and permeabilized (0.2% Triton X-100) for 10 min. Blocking was performed using 5% goat serum (Sigma, G6767-100 mL) in PBS for 15 min and incubated with primary antibody (1% goat serum/PBS) and secondary antibody (PBS) for 1 h and 45 min, respectively, in a wet chamber. The cells were washed after each step in PBS 3×. At last, the cells were washed in dH_2_O, air-dried and mounted using ProLong™ Gold antifade reagent with DAPI (Thermo Fisher Scientific, Waltham, MA, USA).

### 4.14. Fluorescence Microscopy

Image acquisition for pericentriolar satellites localisation rescue assay was performed on a Delta Vision Core microscope using the oil immersion objective (Plan-Apochromat 60× NA 1.42) and filters for DAPI (435/48), FITC (523/36), TRITC (576/89) and Cy5 (632/22). Z-stacks were acquired at a 1024 × 1024 pixels format and Z-steps of 0.2 microns. The localisation of BBS8 to pericentriolar satellites was evaluated using the Fiji ImageJ (2.9.0, github.com/fiji/fiji).

## Figures and Tables

**Figure 1 cells-12-02662-f001:**
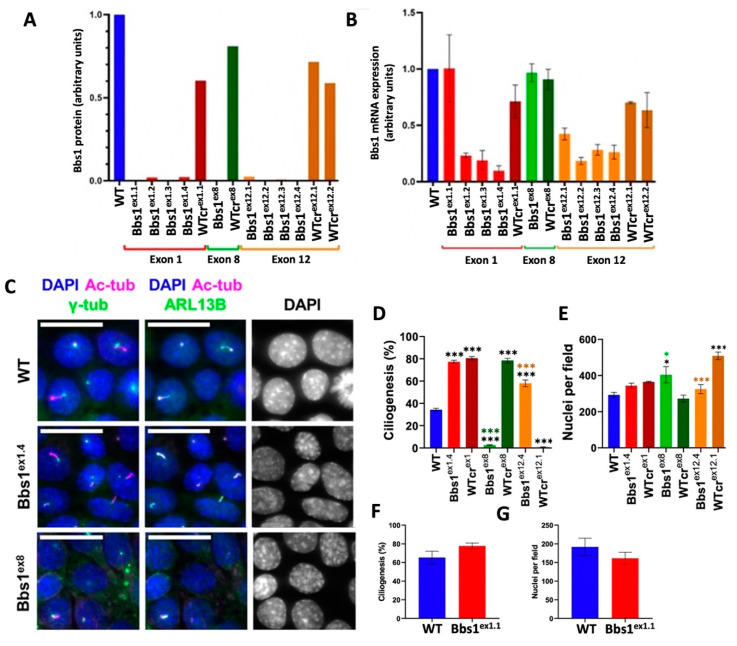
(**A**). Relative levels of the 65 kDa Bbs1 protein. For each cell line, the Bbs1 signal intensity was normalised to that of β-actin. All protein levels (in arbitrary units) are shown relative to the parental WT cell line. Data are for a single experiment. (**B**). Normalised *Bbs1* mRNA expression relative to WT IMCD3 cells. Bars show mean fold change, and error bars show SEM of three independent experiments. (**C**). Representative cilia staining. Cilia were identified as structures which stained for acetylated α-tubulin (ac-tub) and ARL13B. Nuclei were stained with DAPI. Bbs1ex1.1 was able to form cilia. Scale bars represent 20 μm. (**D**). Quantification of ciliogenesis, calculated as the percentage of cells which formed a cilium (co-stained for ac-tub and ARL13B with a γ-tub-positive basal body). Coloured bars show the mean percentage of ciliogenesis, and error bars show SEM of three independent experiments. One-way ANOVA was performed after arcsine square root transformation of proportions of cells with cilia. This indicated a statistically significant difference between the mean proportion of ciliogenesis of cell lines (*p* < 0.0001). Post hoc Bonferroni multiple comparisons tests indicated that all *Bbs1* mutant and WTcr clones differed significantly compared to parental WT (*p*-adj < 0.0001 for all comparisons). Compared to their respective WTcr clones, Bbs1^ex8^ and Bbs1^ex12.4^ differed significantly (*p*-adj < 0.0001 for both comparisons), whereas Bbs1^ex1.4^ and Bbs1^M390R^ did not (*p*-adj > 0.9999 for both comparisons). On the graph, adjusted *p*-values are indicated: * *p* = 0.01 to 0.05; ** *p* = 0.001 to 0.01; *** *p* < 0.001; those in black are relative to WT, and those in colour are relative to the respective WTcr clone. (**E**). Mean number of nuclei per ciliogenesis assay image. Reduced ciliogenesis in Bbs1^ex8^, WTcr^ex12.1^, Bbs1^M390R^ and WTcr^M390R.1^ was not due to decreased cell density. Coloured bars show mean numbers of nuclei per field, and error bars show SEM of three independent experiments. One-way ANOVA indicated a significant difference among means of different cell lines (*p* < 0.0001). Post hoc Bonferroni multiple comparisons tests indicated that, compared to WT, Bbs1^ex8^ (*p*-adj = 0.0038), WTcr^ex12.1^ (*p*-adj < 0.0001) and WTcr^M390R.1^ (*p*-adj = 0.0081) differed significantly, whereas Bbs1^ex1.4^ (*p*-adj > 0.9999), WTcr^ex1^ (*p*-adj > 0.9999), WTcr^ex8^ (*p*-adj = 0.5386), Bbs1^ex12.4^ (*p*-adj > 0.9999) and Bbs1^M390R^ (*p*-adj > 0.9999) did not. Compared to their respective WTcr clones, Bbs1^ex8^ (*p*-adj < 0.0001), Bbs1^ex12.4^ (*p*-adj > 0.0001) and Bbs1^M390R^ (*p*-adj = 0.0133) differed significantly, whereas Bbs1^ex1.4^ did not (*p*-adj > 0.9999). On the graph, adjusted *p*-values are indicated: * *p*-adj = 0.01 to 0.05; ** *p*-adj = 0.001 to 0.01; *** *p*-adj < 0.001; those in black are relative to WT, and those in colour are relative to the respective WTcr clone. (**F**). Quantification of ciliogenesis, calculated as the percentage of cells which formed a cilium (co-stained for ac-tub and ARL13B). Coloured bars show the mean percentage of ciliogenesis, and error bars show SEM of three independent experiments. A paired t-test was performed after arcsine square root transformation of proportions of cells with cilia. Ciliogenesis in Bbs1^ex1.1^ did not differ significantly from that in WT (*p* = 0.0603). (**G**). Mean number of nuclei per ciliogenesis assay image. Coloured bars show mean numbers of nuclei per field, and error bars show SEM of three independent experiments. A paired t-test showed no statistically significant difference between numbers of nuclei per field of Bbs1^ex1.1^ and WT (*p* = 0.1950).

**Figure 2 cells-12-02662-f002:**
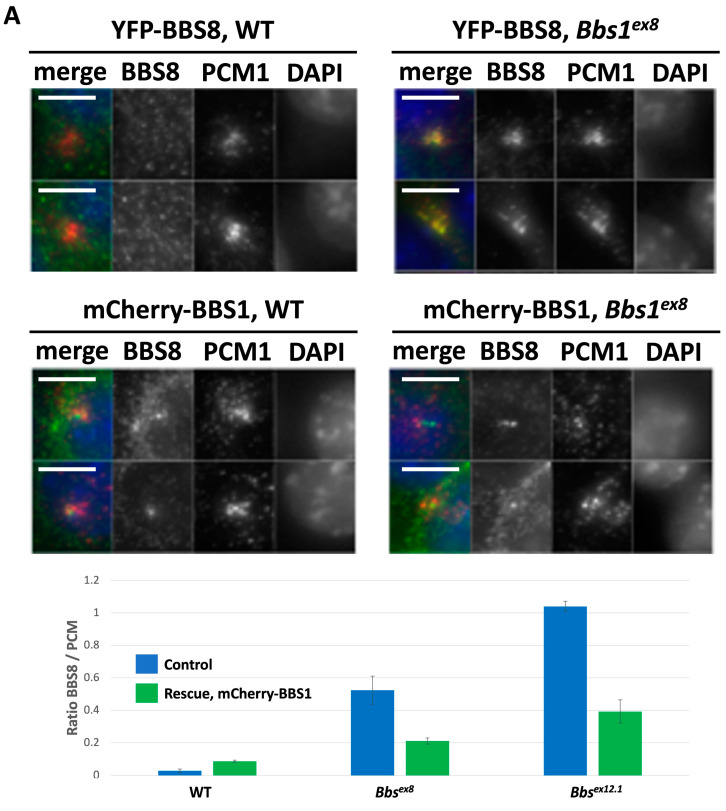
(**A**) Representative images of IMCD3 cell clones of the indicated genotypes transduced with YFP-BBS8, either with or without mCherry-BBS1, showing subcellular localisation of these BBSome components in relation to PCM1. Quantification of the proportion of PCM1 immunofluorescence coinciding with the YFP-BBS8 signal. Scale bars are 2 microns. (**B**) log2 enrichment and peptide intensities comparing affinity purifications for BBS1 wild-type and p.M390R baits versus themselves and a RAF1 control.

**Figure 3 cells-12-02662-f003:**
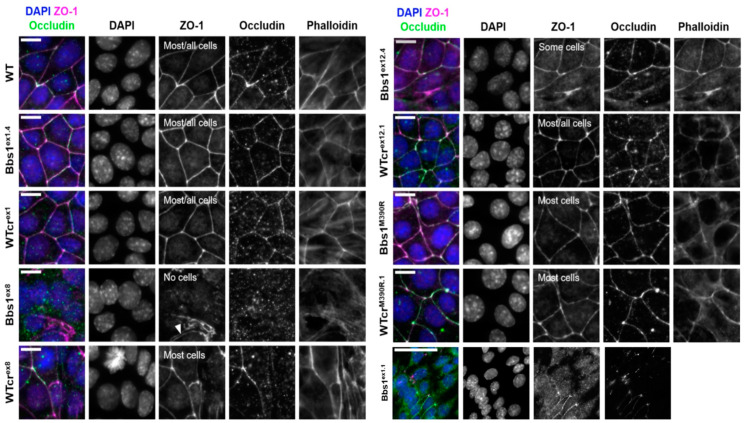
Representative immuno-staining of ZO-1 and occludin in tight junctions. The F-actin cytoskeleton and nuclei were stained with phalloidin and DAPI, respectively. Z-projections of planes containing tight junctions are displayed. Scale bars represent 20 μm. Qualitative assessment of tight junction formation is summarised on the ZO-1 channel. In Bbs1ex8, localisation of ZO-1 to the peripheries of some cells was seen, but there was a failure of cell peripheries to seal together (white arrowhead).

**Figure 4 cells-12-02662-f004:**
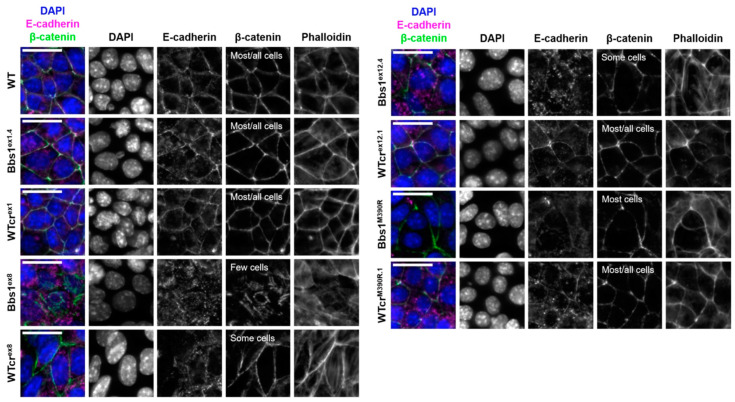
Representative immuno-staining of E-cadherin and β-catenin in adherens junctions. The F-actin cytoskeleton and nuclei were stained with phalloidin and DAPI, respectively. Z-projections of planes containing tight junctions are displayed. Scale bars represent 20 μm. Qualitative assessment of tight junction formation is summarised on the β-catenin channel.

**Figure 5 cells-12-02662-f005:**
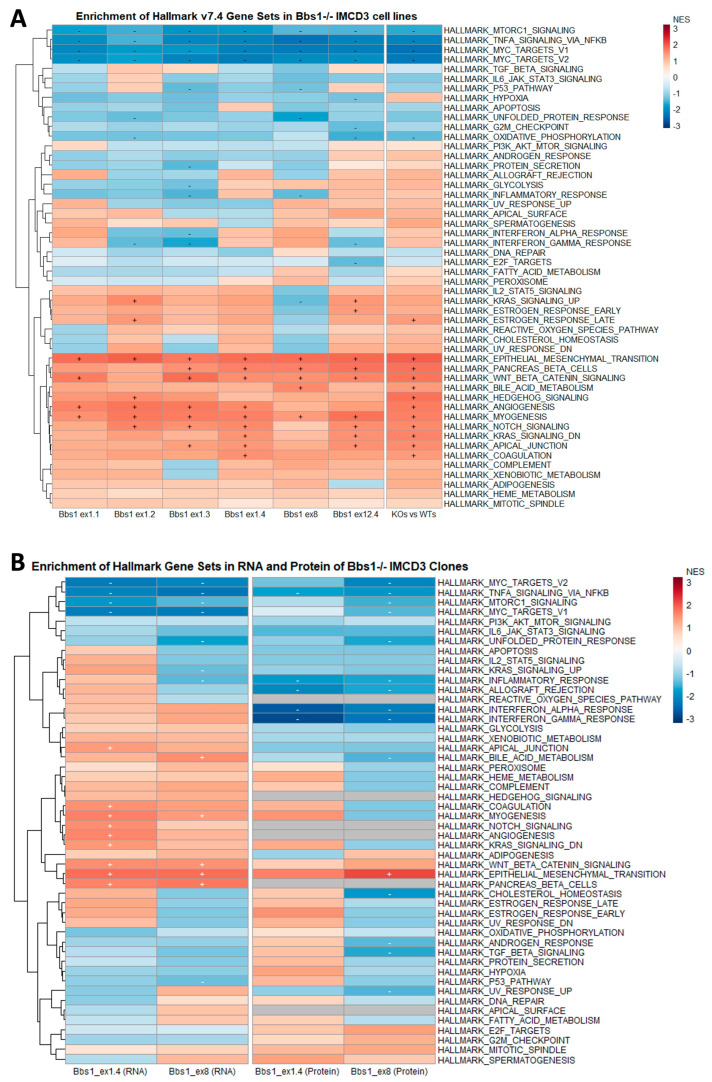
Hierarchical clustering of NES values comparing GSEAs for DEGs observed in 6 IMCD3 *Bbs1*^-/-^ clones vs. Wtcr controls, or all mutant clones versus all Wtcr controls (“KOs vs. WTs”) (**A**), and for DEGs identified in p.M390R patient fibroblasts, mouse hypothalamus and BBS7-depleted HeLa cells (**B**) (‘+’ and ‘-’ indicate statistically significant positive and negative enrichment, respectively. (**C**,**D**) Relative expression of selected mesenchymal genes in *Bbs1*^-/-^ and WtCr clones.

**Figure 6 cells-12-02662-f006:**
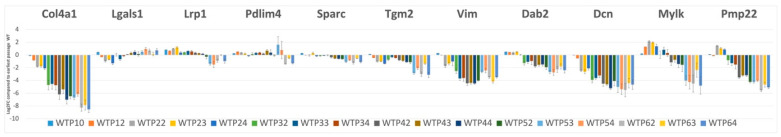
Mesenchymal gene expression in parental (non-clonal) IMCD3 cells (WT) assessed at the indicated passage numbers (P10-64).

## Data Availability

All novel datasets are available on request.

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
