# Peer review of "De-Suppression of Mesenchymal Cell Identities and Variable Phenotypic Outcomes Associated with Knockout of Bbs1"

_cells, 2023, doi:10.3390/cells12222662_

Round 1

Reviewer 1 Report

Comments and Suggestions for Authors

In this article, authors presented a comprehensive study to understand the cellular phenotypes associated with a complete loss of BBS1 protein in IMCD3 cells. They have generated a panel of clonal IMCD3 cell lines with biallelic mutation in Bbs1 to understand the pathophysiology of Bardet-Biedl syndrome. Author have performed a detailed clonal analysis of mutant and control to address the variation among clones and off-target effects that provide valuable insight to the BBS1 function inside the cells. This approach ensures the robustness of the conclusions and provides valuable resources for the scientific community. Here are some comments and suggestions:

1.       The authors mention that three different sgRNAs were designed to target different exons of the Bbs1 gene, but the rationale for targeting specific exons is not discussed. Providing a brief explanation of why these exons were selected for targeting would enhance the reader's understanding.

2.       Has the author observed any correlation between the variations in phenotypes, transcript levels, and gene copy numbers among BBS1 clones?

3.       Line 170, briefly explain pre-BBSome complex. In, Fig 2A, PCM1 staining and YFP-BBS8 localization has been shown. However, it might be beneficial to mention why PCM1 staining is relevant to studying the BBSome.

4.       The use of GSEA to assess clonal differences in the transcriptomes of Bbs1-/- clones is a powerful approach to address the clonal variation. Could you also discuss the clonal variation identified in BBs1 clones in GSEA.

5.       The identification of EMT-related gene expression changes in Bbs1-/- clones is an intriguing finding. It would be helpful to explore the potential mechanisms underlying this link. Are there specific pathways or signaling cascades that might be affected by BBS1 deficiency and contribute to these changes?

6.       In addition to the enrichment of the Epithelial Mesenchymal Transition (EMT) Hallmark gene set, did the authors identify any other cilia-related gene datasets in all BBS1 clones that were positively or negatively enriched?

7.       It would be useful to clarify whether the identified EMT-like changes are unique to BBS1 mutations or if they are a common feature in various ciliopathies. What is the possible implication of enrichment of EMT genes in the bbs1 clones.

8.       Could you comment on the possible reasons behind this clonal variation and its implications for studying BBS1 function.

Overall, this is a valuable study that significantly contributes to our understanding of BBS1 and its role in cellular phenotypes, with the potential to impact the broader field of ciliopathies.

Comments on the Quality of English Language

none

Author Response

In this article, authors presented a comprehensive study to understand the cellular phenotypes associated with a complete loss of BBS1 protein in IMCD3 cells. They have generated a panel of clonal IMCD3 cell lines with biallelic mutation in Bbs1 to understand the pathophysiology of Bardet-Biedl syndrome. Author have performed a detailed clonal analysis of mutant and control to address the variation among clones and off-target effects that provide valuable insight to the BBS1 function inside the cells. This approach ensures the robustness of the conclusions and provides valuable resources for the scientific community. Here are some comments and suggestions:

  1. The authors mention that three different sgRNAs were designed to target different exons of the Bbs1 gene, but the rationale for targeting specific exons is not discussed. Providing a brief explanation of why these exons were selected for targeting would enhance the reader's understanding.

We have added the following to page 4, para 2:

These gRNAs were selected as high-scoring sites that were considered likely to generate null alleles by targeting the translation initiation codon or being located in exons that would generate out-of-frame transcripts should they be skipped.

  1. Has the author observed any correlation between the variations in phenotypes, transcript levels, and gene copy numbers among BBS1 clones?

The following has now been added as part of an additional section within the discussion together with several paragraphs below, page 10, para 3:

We did note that cell clones with most severe phenotypes (ex8, ex1.1) demonstrated resistance to Bbs1 transcript instability (i.e. higher transcript abundance) than other clones. In quantitative transepithelial electrical resistance (TEER) assays, which is a functional assessment of tight junctions, a strong inverse correlation between Bbs1 transcript abundance and TEER was observed (Supplementary Figure 7). We considered that this might reflect a form of genetic compensation, whereby transcript instability is detected by an as-yet-uncharacterised mechanism and leads to upregulation of functionally paralogous genes. However, we were unable to detect such a candidate gene and this interesting possibility will require future investigation.

  1. Line 170, briefly explain pre-BBSome complex. In, Fig 2A, PCM1 staining and YFP-BBS8 localization has been shown. However, it might be beneficial to mention why PCM1 staining is relevant to studying the BBSome.
  2. The use of GSEA to assess clonal differences in the transcriptomes of Bbs1-/- clones is a powerful approach to address the clonal variation. Could you also discuss the clonal variation identified in BBs1 clones in GSEA.

The following has now been added as part of an additional section within the discussion together with several paragraphs below, page 10, para 3:

Visual inspection of the GSEA heatmap data does support a consistent transcriptional signature across all Bbs1-/- clones representing at least 16 gene sets. Although not all of these were statistically significant in all clones, this could reflect statistical power, and so we have certainly focused on the most high-confidence findings. The whole cell proteomics of just two discordant clones also showed that a number of gene sets were commonly to both, and EMT was once again enriched in both clones. Therefore, there may be Bbs1-independent sources of clonal variation, factors of variation that interact with Bbs1 and also differences in cell state, as discussed below.

  1. The identification of EMT-related gene expression changes in Bbs1-/- clones is an intriguing finding. It would be helpful to explore the potential mechanisms underlying this link. Are there specific pathways or signaling cascades that might be affected by BBS1 deficiency and contribute to these changes?

The following has now been added as part of an additional section within the discussion together with several paragraphs below that address other points, page 10, para 3:

Apologies for lack of clarity – in Figure 5A and B, mTORC1, TNFA and MYC targets are negatively enriched in both transcriptomics and proteomic datasets, and WNT/Beta-catenin, Pancreas/Beta-cells, and Myogenesis gene sets are positively enriched in all but one clone. WNT and MYC have been linked to EMT and fibrosis and so could be valuable targets to investigate in future work.

  1. In addition to the enrichment of the Epithelial Mesenchymal Transition (EMT) Hallmark gene set, did the authors identify any other cilia-related gene datasets in all BBS1 clones that were positively or negatively enriched?

The following has now been added as part of an additional section within the discussion together with several paragraphs below, page 10, para 3:

Apologies for lack of clarity – in Figure 5A and B, mTORC1, TNFA and MYC targets are negatively enriched in both transcriptomics and proteomic datasets, and WNT, Pancreas/Beta-cells, and Myogenesis gene sets are positively enriched in all but one clone.

  1. It would be useful to clarify whether the identified EMT-like changes are unique to BBS1 mutations or if they are a common feature in various ciliopathies. What is the possible implication of enrichment of EMT genes in the bbs1 clones.

This is a very interesting and important question. We have tried to comment on this in XXXX, where we discuss neural crest delamination and migration in BBS and also possible relevance to fibrosis. We also noted similar phenotypes reported previously (e.g. IFT140), however, this finding is indeed entirely novel, to our knowledge. We thought it sensible to resist speculating any further.

  1. Could you comment on the possible reasons behind this clonal variation and its implications for studying BBS1 function.

 The following has now been added as part of an additional section within the discussion together with several paragraphs below, page 10, para 3:

Clonal variation could relate to genetic or epigenetic differences, or heterogeneity of cellular state which could cause cells to respond differently to the homogenous experimental conditions in which they were grown. Our WGS did indeed demonstrate genetic differences between clones, but no mutations were shared amongst clones or present in homozygous form in ciliary genes. (We did identify a SNP in Bbs1 present in all clones, and the parental cells). In terms of epigenetics, we would expect to see these differences reflected in transcriptional changes. We did identify discordant gene expression changes when a directed analysis of the severe ex8 and mild ex1.4 clones were compared. However, a wider study will be required to follow up on any relevance of these differences to clonal variation. The finding that a significant proportion of IMCD3 cell clone never form cilia, whereas a typical rate of ciliation of approximately 60% in all other clones, does indeed suggest that different cellular states may be present. Once again, a systematic comparison of panels of wild-type clones with and without ciliogenic potential would need to be undertaken. In our work, for the purposes of rigour, we focused on the EMT phenotype that was consistent across clones.

Overall, this is a valuable study that significantly contributes to our understanding of BBS1 and its role in cellular phenotypes, with the potential to impact the broader field of ciliopathies.

Comments on the Quality of English Language

none

Submission Date

28 September 2023

Date of this review

13 Oct 2023 20:04:40

Reviewer 2 Report

Comments and Suggestions for Authors

The manuscript by Freke et al., provides an analysis of the morphological, functional, and transcriptional changes undergone by multiple models of BBS cells derived from editing the Bbs1 gene. The authors aim to investigate BBS disease pathogenesis using these models (and matched unaltered cells along with patient-derived cells). Using these models in a systemic approach, the authors demonstrate morphological, transcriptional, and functional changes across these models.

Overall, this paper is extremely well-written, the experiments are appropriately planned and executed, and the authors' conclusions are in line with the results presented. 

Regarding the experimental results in the context of BBS cell biology, the authors comprehensively demonstrate changes to BBSome, tight junctions, and adherens junctions. Further, the authors demonstrate a transcriptional EMT phenotype which likely explains and/or complements previous findings in this field. Overall, the results of this paper are an important addition to the BBS cell biology field. 

Additionally, the authors have produced their data in the context of multiple cell lines generated from gene editing and which display varying levels of phenotype. The inclusion of these multiple cell lines is a strength of the paper as it both solidifies their reported data and also demonstrates the inherent variability in gene-edited cell lines. Reviewers and readers can be quite confident in the authors results and interpretations because of the inclusion of these multiple models. 

In summary, this study exactly fits the scope of Cells and serves as an important contribution to the BBS cell biology field. I reccomend acceptance in the present form. 

Author Response

R2

Comments and Suggestions for Authors

The manuscript by Freke et al., provides an analysis of the morphological, functional, and transcriptional changes undergone by multiple models of BBS cells derived from editing the Bbs1 gene. The authors aim to investigate BBS disease pathogenesis using these models (and matched unaltered cells along with patient-derived cells). Using these models in a systemic approach, the authors demonstrate morphological, transcriptional, and functional changes across these models.

Overall, this paper is extremely well-written, the experiments are appropriately planned and executed, and the authors' conclusions are in line with the results presented. 

Regarding the experimental results in the context of BBS cell biology, the authors comprehensively demonstrate changes to BBSome, tight junctions, and adherens junctions. Further, the authors demonstrate a transcriptional EMT phenotype which likely explains and/or complements previous findings in this field. Overall, the results of this paper are an important addition to the BBS cell biology field. 

Additionally, the authors have produced their data in the context of multiple cell lines generated from gene editing and which display varying levels of phenotype. The inclusion of these multiple cell lines is a strength of the paper as it both solidifies their reported data and also demonstrates the inherent variability in gene-edited cell lines. Reviewers and readers can be quite confident in the authors results and interpretations because of the inclusion of these multiple models. 

In summary, this study exactly fits the scope of Cells and serves as an important contribution to the BBS cell biology field. I reccomend acceptance in the present form. 

Submission Date

28 September 2023

Date of this review

12 Oct 2023 17:21:17

Reviewer 3 Report

Comments and Suggestions for Authors

Grace Freke et al submitted their findings under the title "De-suppression of mesenchymal cell identities and variable 2 phenotypic outcomes associated with knockout of Bbs1"

The data is interesting, however, there are few minor comments:

*Add IRB approval number and institute name in the first paragraph of material and methods.

*Total AR BBS types are 22 listed in recent publication (PMID: 37239474 ) and OMIM. 

*Remove "." from the title.

*There are typos, kindly carefully review the MS. eg "Mass spec run". write complete name.

* References should be in same style. The authors wrote "PMID: 420 19688738" in the text, and used "Dobin et al., 2013" and also used "1,2,3", for others.

*gene name should be italic.

*add tools used for statistics in M & M.

Author Response

R3

Comments and Suggestions for Authors

Grace Freke et al submitted their findings under the title "De-suppression of mesenchymal cell identities and variable 2 phenotypic outcomes associated with knockout of Bbs1"

The data is interesting, however, there are few minor comments:

*Add IRB approval number and institute name in the first paragraph of material and methods.

We have now added this, as suggested.

*Total AR BBS types are 22 listed in recent publication (PMID: 37239474 ) and OMIM. 

Thank you the reference has been included.

*Remove "." from the title.

Thank you this has been done.

*There are typos, kindly carefully review the MS. eg "Mass spec run". write complete name.

This has been changed.

* References should be in same style. The authors wrote "PMID: 420 19688738" in the text, and used "Dobin et al., 2013" and also used "1,2,3", for others.

This has been checked.

*gene name should be italic.

This has been checked throughout

*add tools used for statistics in M & M.

This has been added.

Submission Date

28 September 2023

Date of this review

05 Oct 2023 08:46:34

Reviewer 4 Report

Comments and Suggestions for Authors

Perhaps I missed in the text why the phalloidin staining of Bbs1ex1.1 cells is missing in figure 3, but adding it would make this drawing complete

Author Response

R4

Comments and Suggestions for Authors

Perhaps I missed in the text why the phalloidin staining of Bbs1ex1.1 cells is missing in figure 3, but adding it would make this drawing complete

This assay was not done for this clone, as there was time separation for this clone which was made at a later date.

Submission Date

28 September 2023

Date of this review

13 Oct 2023 18:07:04